# Hormonal dynamics and transcriptomic regulatory mechanisms during seed dormancy release in *Aconitum kusnezoffii*

**Yingtong Mu, Kefan Cao, Xiaojie Li\*, Xiaoming Zhang\***

College of Grassland Science/Key Laboratory of Grassland Resources of Ministry of Education, Inner Mongolia Agricultural University, Hohhot, China

\* nnd199511@163.com (XL); 915544393@qq.com (XZ)

## Abstract

*Aconitum kusnezoffii* is an important medicinal plant whose seeds exhibit deep physiological dormancy. To elucidate the endogenous hormonal regulation mechanisms and dynamic gene expression patterns during dormancy release, we conducted a comprehensive analysis integrating hormone quantification and transcriptome sequencing on seed samples subjected to cold stratification for 0, 14, and 42 days. Morphological observations indicated that the embryo development rate remained stable during the first 14 days of stratification but increased significantly after 21 days, displaying a sigmoidal growth pattern. This provided a rationale for delineating the dormancy maintenance, release initiation, and pre-germination phases and for selecting key sampling time points. Hormonal profiling revealed a continuous decline in abscisic acid (ABA) content, accompanied by a gradual increase in germination-promoting hormones such as gibberellins (bioactive GA, reported as $GA_3$-equivalents), indole-3-acetic acid (IAA), brassinolide (BL) and methyl jasmonate (MeJA). The ratios of GA/ABA ($GA_3$-equivalents), IAA/ABA, and zeatin riboside (ZR; ELISA-based total tZR + cZR equivalents)/ABA increased accordingly, suggesting that dormancy release is orchestrated through a synergistic interplay of multiple hormones. Transcriptome analysis yielded 79,251 Unigenes, with differentially expressed genes significantly enriched in pathways related to energy metabolism, hormone signal transduction, and carbohydrate metabolism. STEM clustering identified five representative temporal expression profiles. Further analysis of hormone-related pathways showed that ABA biosynthesis and signaling components were predominantly active at early stages but declined thereafter, whereas GA and IAA biosynthetic and signaling genes were markedly upregulated during the later stages of dormancy release. These findings highlight the close association between embryo development and hormonal dynamics during seed dormancy release in *A. kusnezoffii*, and clarify the stage-specific features of hormone synthesis and signaling

**Data availability statement:** All relevant data are within the manuscript and its Supporting Information files.

**Funding:** The author(s) received no specific funding for this work.

**Competing interests:** The authors have declared that no competing interests exist.

pathways, providing a theoretical basis for understanding dormancy mechanisms and improving artificial germination techniques.

## Introduction

*Aconitum kusnezoffii* Reichb., a perennial herbaceous species of the Ranunculaceae family, is a traditional medicinal plant widely used in Mongolian and Chinese medicine. Its dried tuber, commonly referred to as "Cao Wu," exhibits notable pharmacological properties such as wind-dispelling, dampness-eliminating, and analgesic effects [1]. Currently, the medicinal material is mainly sourced from wild populations, and the increasing depletion of these natural resources has posed significant limitations on its large-scale artificial cultivation [2]. Seeds of *A. kusnezoffii* exhibit distinct morphological and physiological dormancy characteristics [3–5]. Their naturally low germination rate has become a major bottleneck hindering commercial propagation efforts [6].

Seed dormancy is an adaptive strategy that enables plants to withstand unfavorable environmental conditions by establishing an ecological buffer between seed maturation and germination, thus ensuring species survival [7]. International research has long focused on the hormonal regulation and signal transduction networks controlling dormancy. It is now well established that abscisic acid (ABA) plays a key inhibitory role in dormancy maintenance, while gibberellins (GA) act as primary promotive hormones that facilitate dormancy release and germination induction [8,9]. The classic hypothesis involving the GA/ABA ratio has been extensively validated in model and crop species such as *Arabidopsis thaliana* [10], *Triticum aestivum* [11], *Oryza sativa* [12], and *Glycine max* [13]. Additionally, hormonal crosstalk, including antagonistic or synergistic interactions between auxin (IAA), jasmonic acid (JA), brassinosteroids (BR), and the ABA/GA axis, has been identified as a critical regulatory layer determining the timing of dormancy release [14]. In recent years, the roles of non-classical hormones such as methyl jasmonate (MeJA) and zeatin riboside (ZR) in stress-responsive germination and signaling processes have also attracted increasing attention [15–17].

At the molecular level, high-throughput transcriptome technologies have been widely employed to dissect the regulatory networks underlying seed dormancy [16,18]. Differentially expressed gene (DEG) analysis and KEGG enrichment have revealed multiple key pathways, including sugar and lipid metabolism, hormone biosynthesis and signaling, and transcription factor regulation. In species such as *A. thaliana*, *Medicago truncatula*, and *T. aestivum*, genes such as *CYP707A*, *GA2ox*, *DELLA*, *ABI5*, *MYB*, and *bHLH* have been shown to be strongly associated with dormancy status [19–22]. Moreover, dynamic changes in transcription factor families such as *bZIP*, *ERF*, *C2H2*, and *MYB* are considered central regulators in the integration of hormonal signals [23]. However, research on endogenous hormonal changes and molecular regulatory mechanisms during seed dormancy release in non-model medicinal plants—particularly cold-region perennials—is still limited.

As an important medicinal species, research on seed dormancy in *A. kusnezoffii* remains in its infancy. Most prior studies have focused on morphological observations and basic physiological measurements, while the hormonal regulation and underlying molecular mechanisms have yet to be systematically investigated [24]. In this study, we quantified the dynamic changes of six endogenous hormones (ABA, GA$_3$, IAA, MeJA, BR, and ZR) during dormancy release in *A. kusnezoffii* seeds using ELISA, and integrated these data with transcriptome sequencing. We performed comprehensive analyses on differentially expressed genes, hormone biosynthesis and signaling pathways, and transcription factor expression profiles. Furthermore, quantitative real-time PCR (qRT-PCR) was employed to validate the expression patterns of key genes. The objective of this study was to elucidate the hormonal regulatory network involved in seed dormancy release in *A. kusnezoffii* and to provide a theoretical foundation for seedling propagation and sustainable resource utilization.

## Materials and methods

### Plant materials and dormancy-breaking treatment

Seeds of *Aconitum kusnezoffii* Reichb. were collected from the Saihanwula Nature Reserve, Bairin Right Banner, Chifeng City, Inner Mongolia Autonomous Region, China. Healthy seeds with intact seed coats and full shape were selected, soaked in distilled water for 48 h, and mixed with sterilized fine sand at a ratio of 1:3 (w/w). Stratification treatment was conducted at alternating temperatures of −5°C and 4°C every 7 days for a total duration of 56 days. Samples were collected at three critical time points: day 0 (onset of dormancy), day 14 (imbibition stage), and day 42 (radicle emergence stage) for hormone quantification and RNA extraction. Each treatment included three biological replicates, with each biological replicate comprising 30 seeds randomly selected from the bulked seed lot to ensure independent sampling.

### Hormone quantification

Fresh seed samples (1.0 g) were ground to a fine powder in liquid nitrogen using a pre-chilled mortar and pestle and homogenized with 10 mL of ice-cold 80% (v/v) methanol containing 1 mM butylated hydroxytoluene (BHT) as an antioxidant. Polyvinylpolypyrrolidone (PVPP, 1% w/v) was added to remove phenolic compounds. The homogenates were incubated at 4 °C for 12 h with gentle shaking and subsequently centrifuged at 12,000 × g for 15 min at 4 °C. The supernatants were collected and evaporated to dryness under a gentle nitrogen stream at 35 °C. The residues were reconstituted in phosphate-buffered saline (PBS, pH 7.4) and centrifuged again at 12,000 × g for 10 min prior to ELISA analysis. The contents of abscisic acid (ABA), gibberellins (reported as GA$_3$-equivalents), indole-3-acetic acid (IAA), methyl jasmonate (MeJA), brassinolide (BL), and zeatin riboside (ZR) were quantified using enzyme-linked immunosorbent assay kits (Shanghai Enzyme-linked Biotechnology Co., Ltd.). For ZR quantification, the ELISA antibody exhibits cross-reactivity with both trans-zeatin riboside (tZR) and cis-zeatin riboside (cZR); therefore, ZR values are reported as the combined tZR + cZR equivalents. All assays were performed with three independent biological replicates, each measured in triplicate, following the manufacturer's instructions. Absorbance was measured at 450 nm using a microplate spectrophotometer. ELISA was selected for its suitability in high-throughput analysis of small-quantity seed samples; however, it cannot unambiguously distinguish hormone isomers or individual GA species. Accordingly, the obtained values are interpreted as proxies of hormone bioactivity. Future studies will employ targeted LC–MS to enable absolute quantification and precise speciation of free-base cytokinins (tZ/cZ) and bioactive gibberellins (GA$_1$/GA$_4$).

### RNA extraction and transcriptome sequencing

Seed tissues (0.5 g) collected at days 0, 14, and 42 were used for total RNA extraction using the TRIzol reagent. RNA integrity was evaluated using the Agilent 2100 Bioanalyzer, and RNA concentration and purity were measured by NanoDrop 2000. Qualified RNA samples were sent to OE Biotech (Shanghai, China) for library preparation and sequencing on the Illumina NovaSeq platform.

 

## Data processing and differential expression analysis

Raw sequencing data were quality-checked using FastQC and trimmed for adaptors and low-quality reads using Trim Galore to obtain clean reads. De novo assembly was performed using Trinity software under a reference-free condition. Redundant transcripts were removed using CD-HIT to generate a set of representative Unigenes. Functional annotation of Unigenes was conducted by DIAMOND alignment against Nr, SwissProt, KOG, KEGG, and GO databases, while Pfam annotation was performed using HMMER.

Differentially expressed genes (DEGs) were identified using the DESeq2 package in R, with selection criteria set as $|\log_2 \text{FoldChange}| \geq 1$ and FDR < 0.05. KEGG pathway enrichment of DEGs was conducted using the KOBAS tool. Gene expression trend clustering was performed using STEM software to identify significant time-series expression patterns.

## Chemicals and reagents

Commercial enzyme-linked immunosorbent assay (ELISA) kits for abscisic acid (ABA), gibberellin $A_3$ (GA$_3$), indole-3-acetic acid (IAA), methyl jasmonate (MeJA), brassinolide (BL), and zeatin riboside (ZR) were obtained from Shanghai Enzyme-linked Biotechnology Co., Ltd. (China). TRIzol™ Reagent (Invitrogen, USA) was used for RNA extraction. Polyvinylpolypyrrolidone (PVPP; Sigma-Aldrich, USA), butylated hydroxytoluene (BHT; Macklin, China), HPLC-grade methanol (Merck, Germany), and phosphate-buffered saline (PBS) tablets (Solarbio, China) were used during hormone extraction and purification.

## Stage definition

Stage I (day 0, dormant), Stage II (day 14, imbibition), Stage III (day 42, radicle emergence). Morphological landmarks: no embryo growth (Stage I), seed imbibition without radicle protrusion (Stage II), visible radicle emergence (Stage III).

## qRT-PCR Validation

A total of 14 key genes related to hormone biosynthesis, signal transduction, and carbohydrate/lipid metabolism were selected from the transcriptome data for validation. Gene-specific primers were designed (listed in Table 1). Quantitative real-time PCR (qRT-PCR) was performed using the TB Green® Premix Ex Taq™ II kit on an ABI 7500 system, with PlActin

**Table 1. qRT-PCR primer information.**

| NO. | Gene ID | Forward Primer | Reverse Primer |
|---|---|---|---|
| 1 | CYP707A | TCTCGACGGCAACCCAATAT | GGATTCCCAACGACAGATTTGA |
| 2 | PP2C | TCGGTTCCTTCATCTCCCAT | GTGCGGTTCGGATCAAACAA |
| 3 | NCED | AGGTGGGTTGAAGAGCACAG | CCACAGCCCAGACAAGATGT |
| 4 | GA2o | AGCCGCATAAATCACAGCAC | ACCTATCCTGAATCCTGAGTACT |
| 5 | Kao | GAGTTCGTCGGTCTCCTCAG | TTGCTTTGCTGTTCATGGTC |
| 6 | DELLA | TGTGTGGTTGATGTTGGATCT | AGCATGGTCGGTCTTATCATCT |
| 7 | YUCCA | GGAGCAGTGAAGGAGGTGAC | TCTTCCCTTTCCAGCCGTTG |
| 8 | IPT | GAGATCAAGGCCAACACGTCC | CTCCAGCAGGAAGGACCTGAC |
| 9 | ACSL | CTCAGCAGGGCGATTCAGTG | TGCAGCAAAGCAGACTGAAAA |
| 10 | ACS | TGCGGAAGGAGTTGAGAAGT | TGACTCAACCACCTCATGCT |
| 11 | BAM | TGGCAGATGAGTGAATAGCTTC | ACATCTTGCGACTCTCTAATGC |
| 12 | AMY | GGTTGGGCAGGAGAACTTTG | ACCCTTGGTCACTCTCTTGT |
| 13 | PFK | TGTTTGGGAGAGTGGTGGAG | TGGTGAAAATCCTGCAAGCC |
| 14 | GPT | CCTTGGCCGTGTGATTCAAA | TGGCTTAGCTGGTTGTTTGC |

used as the internal control. Each reaction was carried out in a 25 μL volume with three technical replicates. Relative expression levels were calculated using the 2^−ΔΔCt method, and the correlation between RNA-seq and qRT-PCR results was analyzed.

## Data analysis and visualization

Experimental data were statistically analyzed and visualized using Microsoft Excel 2010, R version 4.1.1, Python 3.6, and TBtools software. One-way analysis of variance (ANOVA) was performed to assess the significance of differences, with a threshold of $P < 0.05$ considered statistically significant.

## Results

### Embryo development dynamics during seed dormancy release in *Aconitum kusnezoffii*

To clarify the dynamic progression of dormancy release during stratification, the embryo development rate of *Aconitum kusnezoffii* seeds was systematically monitored across seven time points (0, 7, 14, 21, 28, 35, and 42 days) (Fig 1). Results showed that during the initial 14 days, the embryo development rate remained relatively stable, with values of 23.31%, 23.35%, and 23.84%, respectively, indicating that seed embryos remained in a quiescent state and physiological dormancy had not yet been substantially lifted. By day 21, the rate sharply increased to 38.38%, a significant rise compared with earlier stages ($P < 0.05$), suggesting physiological reactivation of embryos. This upward trend continued, reaching 60.67% on day 28 and further increasing to 83.98% on day 35, eventually peaking at 90.91% on day 42, indicating that most seeds had completed embryo development and entered the germination preparation phase.

Overall, the embryo development rate exhibited a typical sigmoidal (S-shaped) growth curve, with a slow phase during the initial 14 days, followed by rapid acceleration post day 21 and subsequent plateauing. This pattern reflects a gradual transition through three phases: initial dormancy maintenance, physiological activation, and developmental completion. These findings provide key morphological evidence for assessing embryo viability and identifying critical regulatory time points, and they offer a solid rationale for selecting the representative sampling stages (day 0, 14, and 42) used in subsequent hormone quantification and transcriptome analysis.

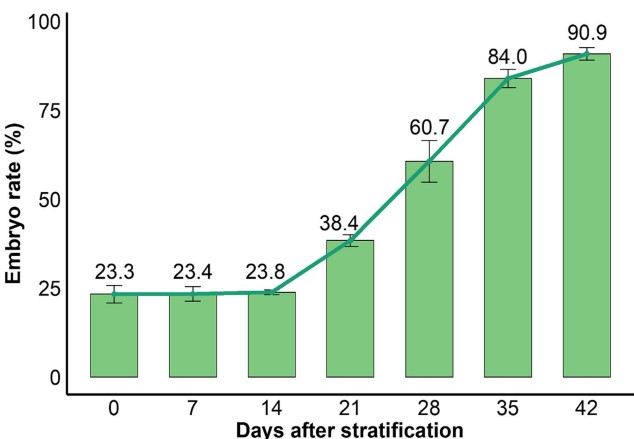

**Fig 1. Dynamic changes in embryo development rate of *Aconitum kusnezoffii* seeds during stratification.**

## Dynamic changes of six endogenous hormones during dormancy release in *Aconitum kusnezoffii* seeds

To elucidate the hormonal regulation associated with seed dormancy release in *Aconitum kusnezoffii*, six endogenous phytohormones—abscisic acid (ABA), gibberellins (bioactive GA, reported as $GA_3$-equivalents), indole-3-acetic acid (IAA), methyl jasmonate (MeJA), brassinolide (BL), and zeatin riboside (ZR)—were quantitatively analyzed at three key developmental stages (Stage I: dormant, Stage II: imbibition, and Stage III: radicle emergence). The results revealed distinct temporal expression patterns among the hormones, reflecting a dynamic balance between inhibitory and promotive signals during the transition from dormancy to germination.

ABA levels declined significantly throughout the stratification process ($P<0.05$), decreasing from 75.55 ng g$^{-1}$ FW at Stage I to 40.43 ng g$^{-1}$ FW at Stage IV, a reduction of 46.48%. This sharp decline underscores the progressive loss of ABA-mediated inhibition, a hallmark of dormancy maintenance (Fig 2a). In contrast, $GA_3$ content increased markedly, rising from 3.02 ng g$^{-1}$ FW at Stage I to a peak of 5.01 ng g$^{-1}$ FW at Stage II ($P<0.05$), then slightly decreasing to 4.07 ng g$^{-1}$ FW at Stage IV, though remaining significantly higher than the initial level ($P<0.05$). These dynamics suggest that $GA_3$ may facilitate embryonic reactivation and metabolic acceleration during early dormancy release (Fig 2b).

IAA exhibited a steady upward trend, increasing from 23.56 ng g$^{-1}$ FW to ng g$^{-1}$ FW ($P<0.05$), indicating an enhanced role in promoting cell division and hypocotyl elongation during the latter phases of dormancy alleviation (Fig 2c). MeJA levels also rose significantly during imbibition ($P<0.05$), from 12.50 ng g$^{-1}$ FW to 25.55 ng g$^{-1}$ FW, and remained elevated at Stage IV (24.69 ng g$^{-1}$ FW), implying its involvement in stress signal transduction and cross-talk regulation during dormancy release (Fig 2d).

BR content remained stable (~2.6–2.7 ng g$^{-1}$ FW) during Stages I and II but significantly increased to 3.22 ng g$^{-1}$ FW at Stage IV ($P<0.05$), suggesting a potential role in initiating germination through cell wall loosening and expansion (Fig 2e). In contrast, ZR displayed a transient peak at Stage II (from 3.14 to 4.35 ng g$^{-1}$ FW, $P<0.05$), followed by a decline to 2.80 ng/g·FW at Stage IV ($P<0.05$), indicating its major involvement in the activation of cell division during the imbibition phase (Fig 2f).

Collectively, these results demonstrate that seed dormancy release in *A. kusnezoffii* is accompanied by a coordinated hormonal transition characterized by a sustained decrease in ABA and concurrent increases in $GA_3$, IAA, and MeJA. The evolving ratios of $GA_3$/ABA, IAA/ABA, and ZR/ABA reflect a finely tuned antagonistic and synergistic regulatory network that orchestrates the progression from dormancy toward germination readiness.

## Changes in the ratios of key endogenous hormones during dormancy release in *Aconitum kusnezoffii* seeds

During the dormancy release process of *Aconitum kusnezoffii* seeds, the ratios of $GA_3$/ABA, IAA/ABA, and ZR/ABA exhibited significant dynamic changes, highlighting the antagonistic interactions among hormones that regulate the transition from dormancy to germination readiness. These hormonal ratios serve as integrative indicators of internal physiological shifts and reflect key regulatory mechanisms underlying dormancy alleviation.

The $GA_3$/ABA ratio increased markedly from 0.040 at Stage I to 0.084 at Stage II ($P<0.05$), and further to 0.101 at Stage IV, although the latter increment was not statistically significant ($P>0.05$). This pattern suggests that the antagonistic effect of $GA_3$ on ABA is rapidly enhanced during the imbibition phase and subsequently stabilized, playing a critical role in the initiation of dormancy release (Fig 3a).

The IAA/ABA ratio showed a consistent upward trend across all three stages, rising from 0.312 at Stage I to 0.572 at Stage II, and reaching 0.936 at Stage IV, with significant differences between each stage ($P<0.05$). This progressive increase indicates that the relative dominance of IAA over ABA is steadily reinforced during dormancy release, likely promoting embryonic cell division and elongation as seeds prepare for germination (Fig 3b).

In contrast, the ZR/ABA ratio followed a "rise-then-decline" trajectory, increasing from 0.042 at Stage I to a peak of 0.073 at Stage II ($P<0.05$), before slightly decreasing to 0.069 at Stage IV, which remained significantly higher than the

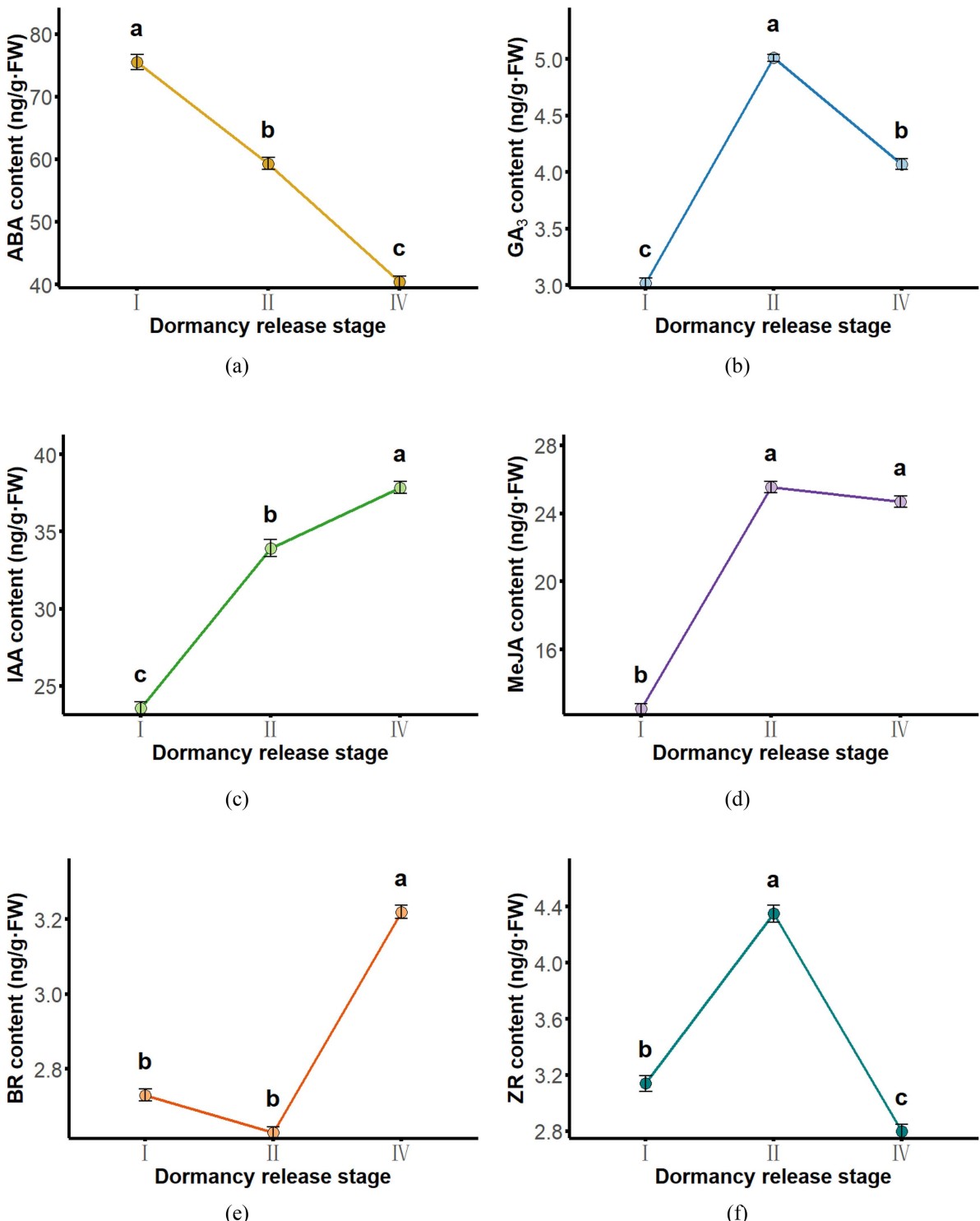

**Fig 2. Dynamic changes in the contents of six endogenous hormones during seed dormancy release in *Aconitum kusnezoffii*: (a) Abscisic acid (ABA); (b) Gibberellin A₃ (GA₃); (c)Indole-3-acetic acid (IAA); (d) Methyl jasmonate (MeJA); (e) Brassinosteroid (BL); (f) Zeatin riboside (ZR; reported as the sum of trans- and cis-isomers, tZR+cZR).** Each value represents the mean ± standard deviation (n = 3). Different lowercase letters indicate statistically significant differences among stages ($p < 0.05$).

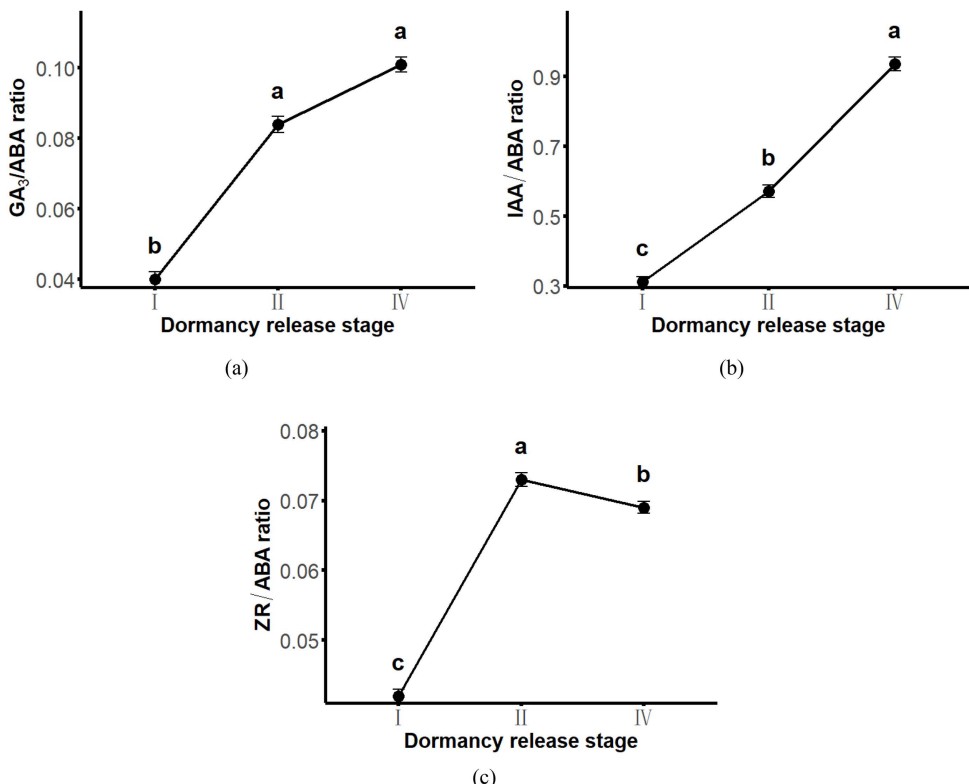

**Fig 3. Dynamic changes in the ratios of key endogenous hormones to ABA during seed dormancy release in *Aconitum kusnezoffii*: (a) GA₃/ABA ratio; (b) IAA/ABA ratio; (c) ZR/ABA ratio.** Each value represents the mean ± standard deviation (n = 3). Different lowercase letters indicate statistically significant differences among stages ($p < 0.05$).

initial value ($P < 0.05$). This suggests that ZR may primarily function during the imbibition phase to activate cell division and meristematic activity, but its regulatory influence diminishes as seeds transition into the final stage of dormancy release (Fig 3c).

Overall, the temporal modulation of these hormone ratios reflects a complex network of hormonal cross-talk, wherein declining ABA levels are counterbalanced by rising promotive hormones (GA₃, IAA, and ZR), orchestrating a gradual shift from dormancy maintenance to germination competence in *A. kusnezoffii* seeds.

### Transcriptomic analysis of *Aconitum kusnezoffii* seeds during dormancy release

**Transcriptome sequencing quality and functional annotation of *Aconitum kusnezoffii* seeds.** To comprehensively explore the transcriptional response mechanisms underlying seed dormancy release in *Aconitum kusnezoffii*, a de novo transcriptome sequencing and annotation was performed on seed samples collected at different stages. Quality statistics of the sequencing data (Table 2) showed that all samples had Q30 values exceeding 92.18%, and a total of 67.03 Gb of clean data were obtained, indicating high-quality library construction and a solid foundation for downstream analysis.

Clean reads were assembled using Trinity software, generating 79,251 transcripts longer than 200 bp. The longest isoforms were retained to construct a representative set of Unigenes for subsequent annotation and differential expression analysis.

**Table 2. Statistics of sequencing data of *Aconitum kusnezoffii*.**

| Sample | Clean reads | Clean bases | GC Content | % ≥ Q30 |
|---|---|---|---|---|
| T1-1 | 50.40M | 7.33G | 47.58% | 92.18% |
| T1-2 | 49.72M | 7.25G | 47.58% | 93.58% |
| T1-3 | 50.35M | 7.35G | 47.61% | 93.85% |
| T2-1 | 47.26M | 6.89G | 47.55% | 93.72% |
| T2-2 | 46.60M | 6.80G | 47.60% | 94.37% |
| T2-3 | 49.42M | 7.20G | 47.52% | 93.84% |
| T3-1 | 49.11M | 7.13G | 45.95% | 94.38% |
| T3-2 | 48.90M | 7.10G | 45.92% | 94.74% |
| T3-3 | 48.56M | 7.03G | 45.88% | 94.41% |

For functional annotation, the Unigenes were compared against several public databases. As a result, 52,317 Unigenes (66.01%) were successfully annotated in the Nr database; 47,116 (59.45%) in eggNOG; 37,590 (47.43%) in Swiss-Prot; 33,070 (41.73%) in Pfam; 30,110 (37.99%) in KOG/COG; 33,447 (42.20%) in GO; and 14,228 Unigenes (17.95%) were annotated in the KEGG database. These results suggest that although *A. kusnezoffii* is a non-model medicinal plant with a substantial proportion of potentially novel genes, the high annotation rates across multiple databases support the robustness of transcriptome assembly and provide a reliable basis for pathway enrichment and regulatory network analysis at the molecular level.

**Transcriptome sequencing quality and functional annotation of *Aconitum kusnezoffii* seeds.** Significant transcriptional changes were observed between different stages of dormancy release in *Aconitum kusnezoffii* seeds. The T1_vs_T3 comparison group exhibited the largest number of differentially expressed genes (DEGs), with a total of 38,250 genes, including 20,985 upregulated and 17,265 downregulated genes. This indicates a substantial transcriptional reprogramming from the initial dormant stage to the late dormancy-released stage. In the T2_vs_T3 group, 28,155 DEGs were identified, comprising 11,728 upregulated and 16,427 downregulated genes, suggesting that major transcriptional adjustments continue during the transition from the imbibition phase to germination readiness. In contrast, the T1_vs_T2 group showed fewer DEGs (16,100 in total), with 13,119 upregulated and 2,981 downregulated, indicating that early dormancy release is characterized mainly by activation of gene expression (Fig 4a).

Among the three pairwise comparisons, a total of 1,756 DEGs were shared across all groups, which may represent core regulatory genes consistently involved throughout the dormancy release process. The T1_vs_T3 comparison group had the highest number of uniquely expressed DEGs (5,171), significantly more than T1_vs_T2 (855) and T2_vs_T3 (4,511), suggesting that the transition from dormancy to germination initiation involves many distinct regulatory genes. Furthermore, 11,462 DEGs were commonly shared between the T1_vs_T2 and T1_vs_T3 groups, highlighting the imbibition phase as a key inflection point in transcriptional regulation, with numerous genes responding dynamically across multiple stages (Fig 4b).

**KEGG enrichment analysis of differentially expressed genes during dormancy release in *Aconitum kusnezoffii*.** The functional enrichment analysis of differentially expressed genes (DEGs) at various stages of dormancy release in *Aconitum kusnezoffii* revealed pronounced temporal specificity in pathway activation. In the T1_vs_T2 comparison, DEGs were significantly enriched in metabolic pathways such as "Glycolysis/Gluconeogenesis," "Citrate cycle (TCA cycle)," "Fatty acid degradation," "Oxidative phosphorylation," "Starch and sucrose metabolism," and "Ribosome." These results indicate that early in dormancy release, seed metabolism is primarily sustained through basic carbon metabolism and energy generation processes (Fig 5a).

In contrast, the T1_vs_T3 group showed broader pathway enrichment. DEGs were involved in pathways including "Ribosome," "Biosynthesis of unsaturated fatty acids," "Flavonoid biosynthesis," "Phenylpropanoid metabolism,"

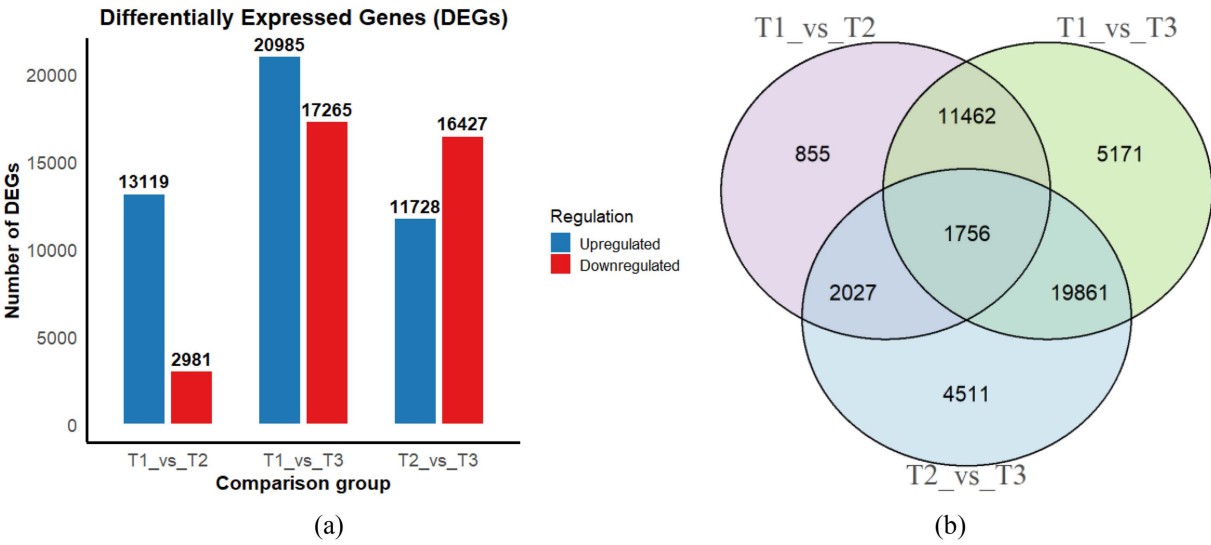

**Fig 4. Summary of differentially expressed genes (DEGs) identified during seed dormancy release in *Aconitum kusnezoffii*: (a) Number of upregulated and downregulated DEGs in each comparison group (T1_vs_T2, T1_vs_T3, T2_vs_T3); (b) Venn diagram showing the overlap of DEGs among the three comparison groups.**

"Photosynthesis," and "Starch and sucrose metabolism." This suggests that during the later stages of dormancy release, there is a marked upregulation of genes related not only to protein synthesis and metabolic activity, but also to plant-specific signal transduction and energy conversion processes. These changes likely contribute to the physiological shift from dormancy to germination readiness (Fig 5b).

The T2_vs_T3 comparison revealed enrichment of DEGs in pathways such as "Protein processing in endoplasmic reticulum," "Proteasome," "Amino acid metabolism," "Carbohydrate metabolism," and "Fatty acid degradation." This implies that metabolic reprogramming at this stage is dominated by active protein synthesis and post-translational modification, along with enhanced regulation of energy and carbon flux, likely supporting embryonic structural reorganization and functional activation (Fig 5c).

Collectively, these results illustrate a dynamic shift in enriched KEGG pathways from core metabolic processes toward more complex regulatory and biosynthetic networks. This transition underscores the multi-layered metabolic and signaling reconfiguration that underlies seed dormancy release and the progression toward germination in *A. kusnezoffii*.

**Differential gene expression trend analysis of *Aconitum kusnezoffii* seed transcriptomes during dormancy release.** To further investigate the temporal dynamic changes of differentially expressed genes (DEGs) during dormancy release in *Aconitum kusnezoffii* seeds, we performed clustering analysis of the full set of DEGs using the Short Time-series Expression Miner (STEM) tool. The analysis identified several statistically significant expression trend modules ($P<0.05$), with clusters 1, 2, 4, 5, and 7 showing marked enrichment, representing dominant expression patterns at different stages and their potential biological functions.

Further KEGG enrichment analysis ($P<0.05$) was conducted for the DEGs within these five trend clusters. The results revealed clear functional assignments for each enriched module. Cluster 2 contained 13,211 genes, which exhibited a significant increase in expression during the early dormancy phase, followed by a gradual decrease. This cluster was significantly enriched in energy metabolism pathways such as oxidative phosphorylation, acetylation and dicarboxylate metabolism, citrate cycle (TCA cycle), fatty acid degradation, glycolysis/gluconeogenesis, and pyruvate metabolism, suggesting that these metabolic processes remain active during dormancy.

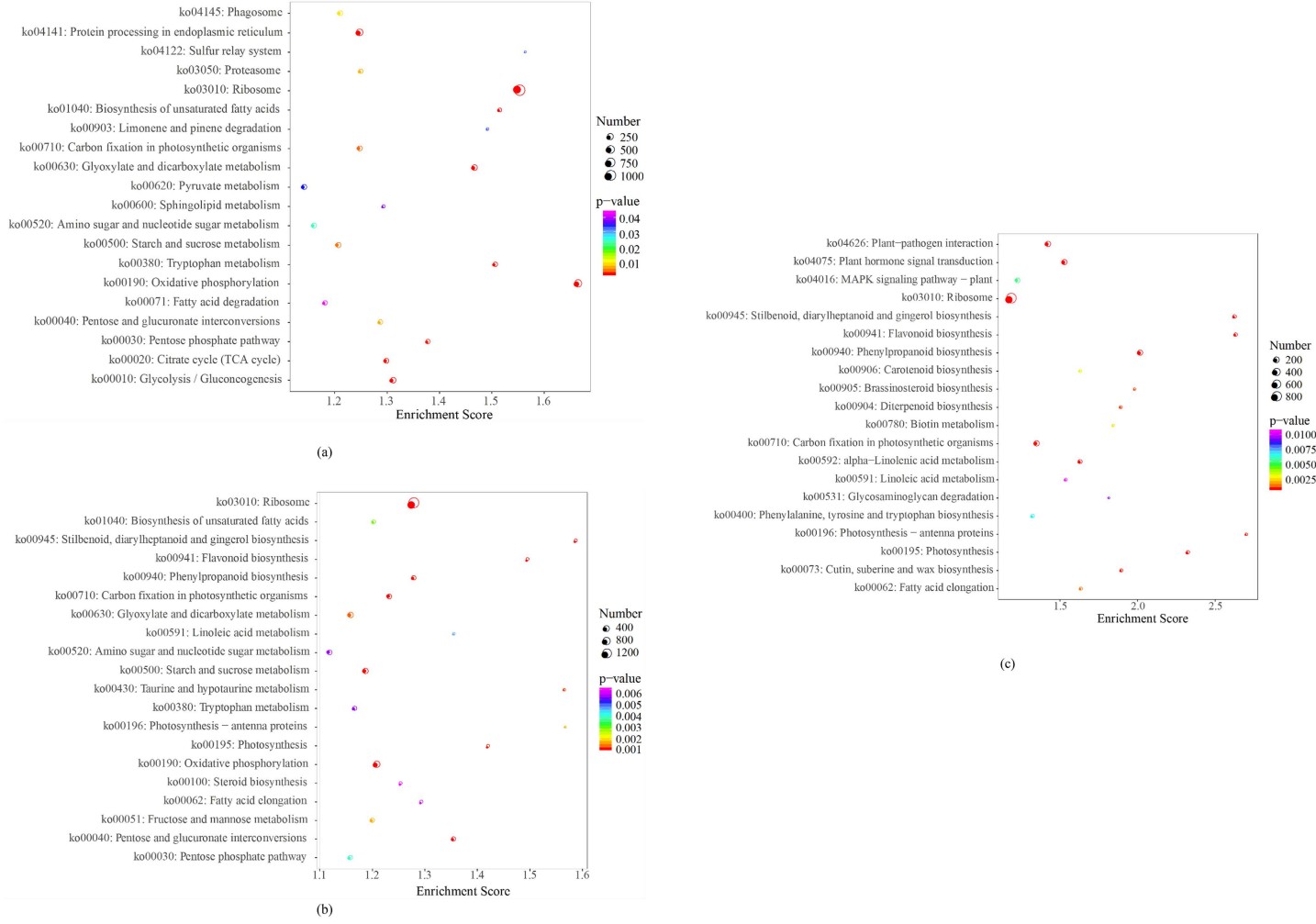

**Fig 5. KEGG enrichment analysis of differentially expressed genes (DEGs) during seed do.**

Cluster 4, consisting of 9,160 genes, was predominantly involved in amino acid metabolism and linoleic acid metabolism, with peak expression levels during the imbibition phase. This suggests an enhancement of cellular metabolic activity during this stage.

Cluster 7, which included 8,456 genes, displayed a continuous upward trend in expression, with significant enrichment in pathways related to plant hormone signal transduction, phenylpropanoid biosynthesis, starch and sucrose metabolism, and pentose and glucuronate interconversions. These findings reflect significant activation of hormone regulation and energy material remodeling in the late dormancy phase.

Cluster 5 contained 5,345 genes, showing the most prominent peak expression during imbibition. It was significantly enriched in pathways such as mRNA surveillance, ribosome biogenesis, nucleotide excision repair, ubiquitin-mediated proteolysis, basic transcription factors, and RNA degradation. This suggests that this phase is crucial for the reactivation and regulation of molecular-level transcription and translation activities.

Cluster 1, with 3,597 genes, showed slightly higher expression during Stage II. This cluster was mainly associated with cytokinin biosynthesis and fructose and mannitol metabolism, which are likely linked to early seedling growth and energy regulation associated with hormone activation in the early germination phase (Fig 6).

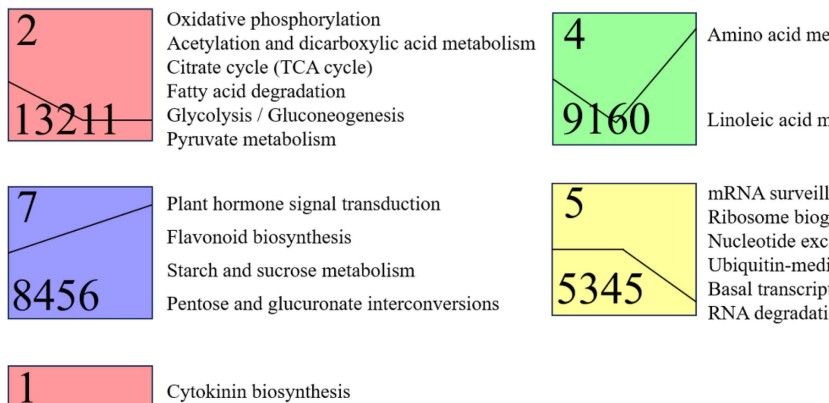

**Fig 6. KEGG pathway enrichment analysis of differentially expressed genes (DEGs) from five significant expression profiles identified by STEM clustering in *Aconitum kusnezoffii* seeds during dormancy release.**

### Expression patterns of hormone-related genes during dormancy release in *Aconitum kusnezoffii* seeds

In this study, dynamic changes in the contents of six phytohormones were monitored to evaluate their potential regulatory roles during the dormancy release of *Aconitum kusnezoffii* seeds. However, annotation of differentially expressed genes (DEGs) revealed that only a limited number of key genes were associated with the MeJA, BR, and ZR signaling pathways, and the relevant pathway information remains partially annotated in current databases, making it difficult to conduct a systematic analysis of expression trends. In contrast, three classical hormones—ABA, GA, and IAA—were significantly enriched in both biosynthesis and signal transduction pathways, with notable expression changes. These hormones have also been extensively studied for their physiological roles in regulating seed dormancy and germination. Therefore, this section focuses on the expression dynamics of genes related to ABA, GA, and IAA during different stages of dormancy release, aiming to elucidate their stage-specific regulatory mechanisms.

**Expression pattern analysis of ABA biosynthesis, catabolism, and signal transduction pathways.** Previous studies have shown that seed dormancy and germination are regulated by the coordinated actions of multiple hormones, with ABA playing a central inhibitory role in the establishment and maintenance of dormancy. To elucidate the regulatory mechanism of ABA during dormancy release in *Aconitum kusnezoffii* seeds, we systematically analyzed ABA-related DEGs based on transcriptomic data, covering three major aspects: biosynthesis, catabolism, and signal transduction.

ABA biosynthesis in plants primarily proceeds via the $C_{40}$ carotenoid pathway. We identified three DEGs encoding *9-cis-epoxycarotenoid dioxygenase* (*NCED*), a rate-limiting enzyme in ABA biosynthesis. These genes were highly expressed at stage T1, slightly decreased at T2, and significantly downregulated at T3, indicating that ABA biosynthesis activity is suppressed during the later stages of dormancy release. Additionally, two DEGs encoding *abscisic aldehyde oxidase* (*AAO3*) showed markedly increased expression during the radicle protrusion stage (T3), contributing to the final step in ABA production from intermediate precursors.

In the catabolic pathway, four DEGs encoding *ABA 8′-hydroxylase* (*CYP707A*) were significantly upregulated at T3, suggesting that the reduction of ABA content is primarily achieved through enhanced degradation. This expression pattern is consistent with the physiological data showing a significant decrease in ABA content at T3, further confirming the link between gene expression regulation and ABA dynamics.

Genes involved in the ABA signal transduction pathway also exhibited stage-specific expression patterns. Five DEGs encoding *PYL* (ABA receptors) were highly expressed at T1 and gradually decreased thereafter, indicating that ABA perception is more active in the early stages of dormancy. Eleven DEGs encoding *PP2C* (protein phosphatases) displayed mixed expression trends but showed an overall upregulation from T2 to T3, suggesting sustained negative regulation of ABA signaling. Four DEGs encoding *SnRK2* (serine/threonine protein kinases) were consistently upregulated throughout dormancy release, implying a positive role in signal activation. Notably, four DEGs encoding *ABF* (ABA-responsive transcription factors) were significantly downregulated at T3, possibly due to upstream regulation by *SnRK2* and feedback inhibition resulting from reduced ABA levels (Fig 7).

**Expression pattern analysis of GA biosynthesis, catabolism, and signal transduction pathways.** Gibberellins (GA) are classical phytohormones known to promote seed germination and play a pivotal role during the dormancy release process. To elucidate the molecular mechanisms underlying GA-mediated regulation in *Aconitum kusnezoffii* seeds, we investigated the differentially expressed genes (DEGs) involved in GA biosynthesis, degradation, and signal transduction pathways.

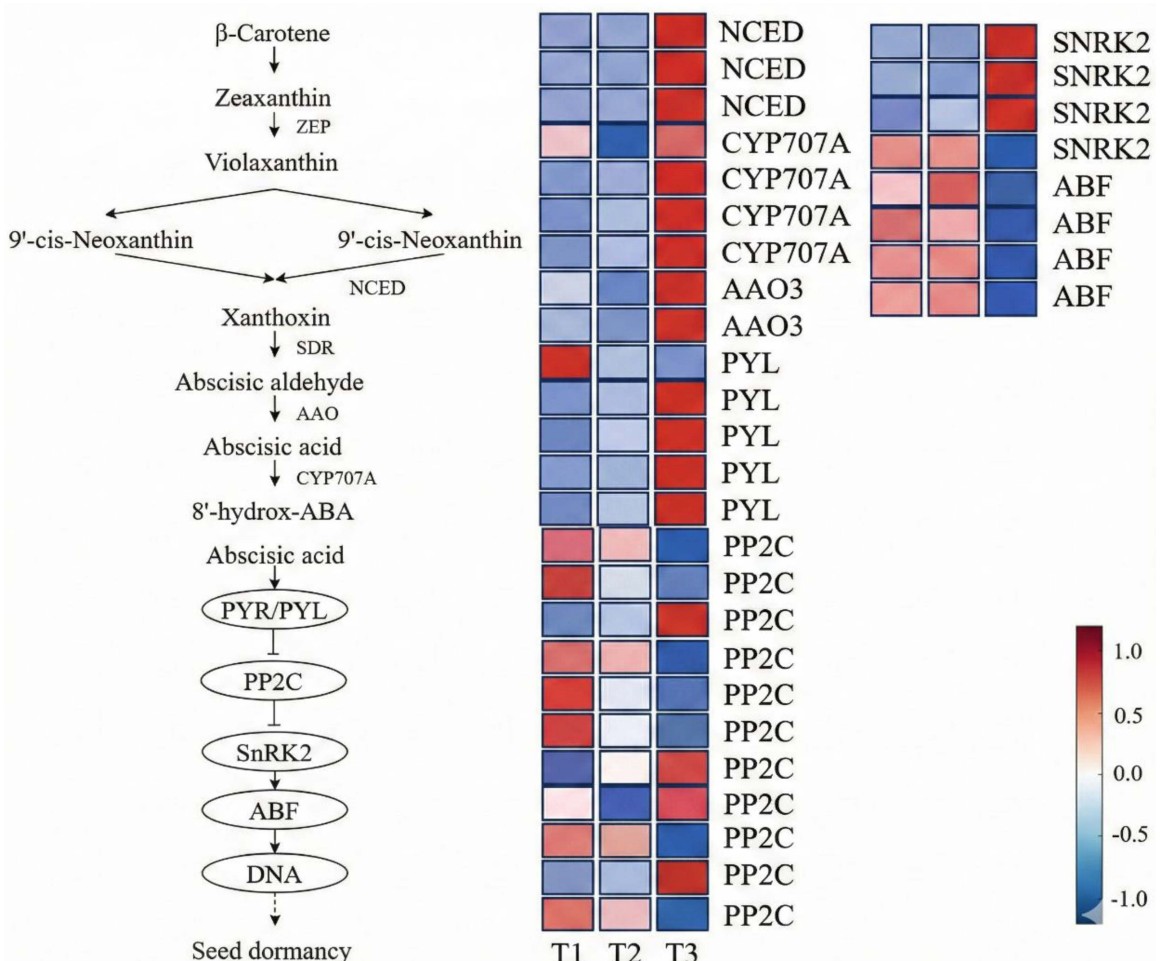

**Fig 7. Expression patterns of key genes involved in the ABA biosynthesis, metabolism, and signal transduction pathways during seed dormancy release in *Aconitum kusnezoffii*.**

In the GA biosynthesis pathway, two DEGs encoding *kaurenoic acid oxidase* (*KAO*) were identified, both of which exhibited continuous upregulation from stage T1 to T3. As a key enzyme responsible for the conversion of ent-kaurenoic acid into bioactive GA precursors, the enhanced expression of *KAO* indicates elevated biosynthetic activity and contributes to the accumulation of GA during dormancy release. In contrast, five DEGs encoding *gibberellin 2-oxidase* (*GA2ox*, EC 1.14.11.13), which catalyze the deactivation of bioactive GA, were significantly downregulated throughout the stratification period. This coordinated regulation—upregulation of *KAO* and suppression of *GA2ox*—suggests a dual mechanism promoting GA accumulation prior to germination.

Within the GA signal transduction cascade, DELLA proteins function as key negative regulators. Three DEGs encoding *DELLA* were significantly downregulated at T3, implying that GA promotes germination by alleviating DELLA-mediated transcriptional repression. Conversely, six DEGs encoding the GA receptor *GID1* and the F-box protein *GID2* were markedly upregulated during T2 and T3 stages, reflecting enhanced sensitivity to GA signals and accelerated degradation of *DELLA* repressors. These changes collectively facilitate the activation of downstream transcription factors, which in turn initiate the expression of genes involved in embryo axis elongation and seed germination.

In summary, *A. kusnezoffii* seeds exhibit a coordinated shift during dormancy release characterized by enhanced GA biosynthesis, suppressed GA deactivation, and activation of the GA signaling pathway. The downregulation of *GA2ox* and upregulation of *KAO* together contribute to elevated GA levels, while concurrent downregulation of *DELLA* and upregulation of *GID1*/*GID2* potentiate signal transmission. These findings demonstrate that the GA signaling module serves as a crucial hormonal switch facilitating the transition from dormancy to germination (Fig 8).

**Expression pattern analysis of IAA biosynthesis and signal transduction pathways.** Indole-3-acetic acid (IAA) is a key hormone in plants that regulates cell elongation and organ development, and its role in seed dormancy release has gained increasing attention. IAA biosynthesis primarily occurs via three pathways: the tryptamine pathway, the indole-3-pyruvic acid (IPA) pathway, and the indole-3-acetonitrile pathway. Among these, the IPA pathway is the most studied and conserved biosynthesis route in plants. In this study, *Aconitum kusnezoffii* seeds predominantly utilized the IPA pathway for IAA synthesis during dormancy release.

In this pathway, three differentially expressed genes (DEGs) encoding *TAA1* (IPA deaminase) were identified, with two genes significantly upregulated at Stage T3 (radicle emergence) and one showing a downregulation trend, indicating potential functional differentiation of *TAA1* across different stages. Two DEGs encoding *YUCCA* (indole-3-acetic acid monooxygenase) showed continuous upregulation throughout the dormancy release process, suggesting that the conversion efficiency from IPA to IAA progressively increased over time, enhancing IAA biosynthesis activity.

In the IAA signal transduction pathway, seven DEGs encoding *AUX1* (auxin influx carrier) were identified, with several genes upregulated during Stages T2 and T3, reflecting enhanced IAA transport capacity across cell membranes. Nine DEGs from the *TIR1* family of F-box proteins were all downregulated at T3, indicating that the negative regulatory role of *TIR1* in signal sensing is suppressed, potentially enhancing downstream signal transduction. Fourteen DEGs encoding *AUX/IAA* transcriptional repressors were significantly upregulated at T3, suggesting that the IAA response network becomes actively engaged during the later stages of dormancy release. Additionally, eight DEGs encoding *GH3* (Gretchen Hagen 3) enzymes were upregulated at T3, which are involved in the conjugation of active IAA with amino acids, playing a key role in IAA metabolic inactivation and providing negative feedback regulation on IAA levels. Three *SAUR* (small auxin-up RNA) genes exhibited different expression patterns, with some showing upregulation at T3, indicating that this class of rapid response elements plays a time-specific role in regulating cell expansion (Fig 9).

Overall, the expression of IAA-related genes during the dormancy release process in *A. kusnezoffii* seeds followed a staged pattern of "synthesis enhancement—transport activation—response initiation." The enhanced activity of the IPA pathway, along with the concurrent upregulation of transcription factors and response genes in the signal transduction module, suggests that IAA not only regulates cell elongation and division but may also drive root primordia formation and hypocotyl growth prior to germination. IAA thus serves as a critical growth-promoting signal during the dormancy release process.

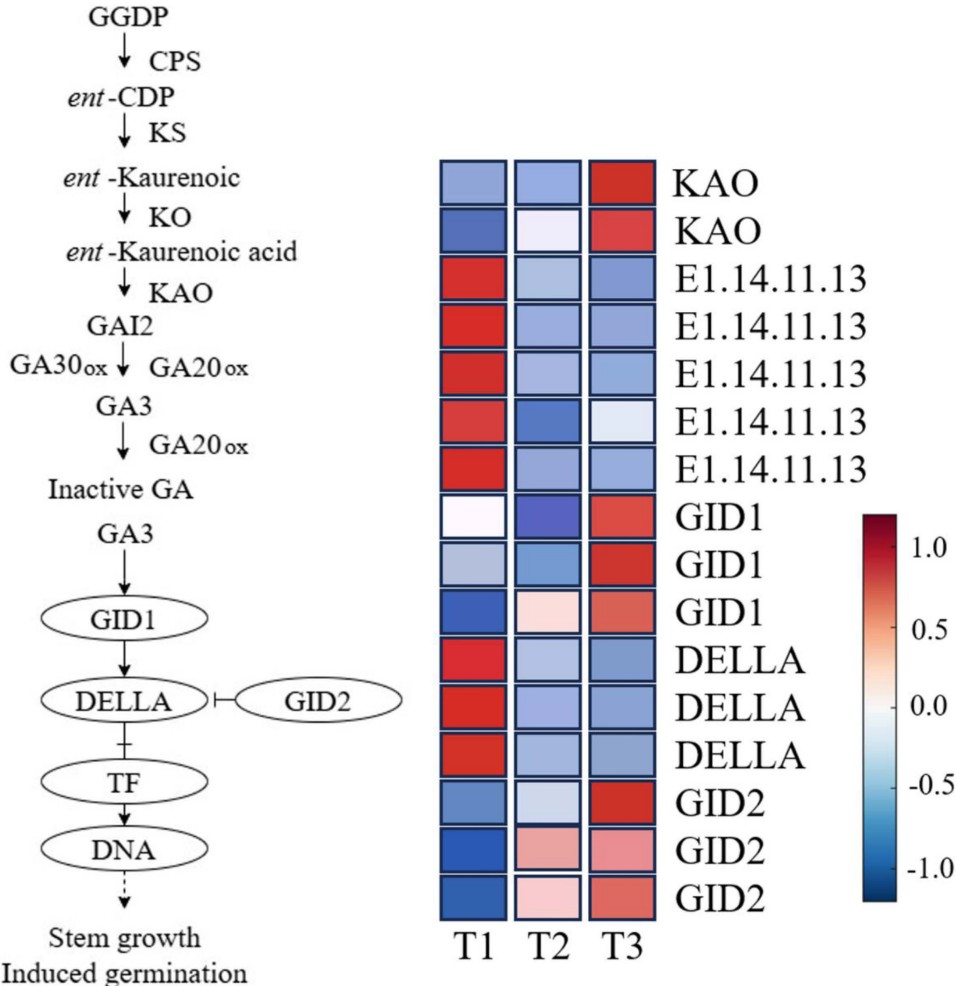

**Fig 8. Expression profiles of key genes involved in GA biosynthesis, deactivation, and signal transduction during seed dormancy release in** *Aconitum kusnezoffii*.

### q-PCR and RNA-seq validation of key gene expression

To validate the reliability of the transcriptome sequencing results, 14 differentially expressed genes (DEGs) related to hormone signaling were randomly selected for qRT-PCR expression validation in this study.

The qRT-PCR results showed that the expression trends of all target genes were generally consistent with the RNA-seq data. Several key genes exhibited significant changes in expression during the dormancy release process. The ABA-related gene *CYP707A* showed increased expression in qRT-PCR, consistent with its upregulation observed in the transcriptome. Similarly, genes involved in ABA signaling, such as *PP2C* and *NCED*, also exhibited high expression levels. For GA signal regulation, genes *GA2ox* and *KAO* showed a decreasing and increasing trend in qPCR, respectively, aligning with the expression patterns observed in the RNA-seq data that suggest GA's role in promoting germination.

Genes involved in IAA and ZR biosynthesis, including *YUCCA* and *IPT*, were significantly upregulated in qRT-PCR, supporting the physiological data showing elevated IAA and ZR contents. Moreover, genes associated with carbohydrate metabolism, such as *AMY*, *PFK*, and *GPT*, were also highly expressed in qPCR, indicating their potential involvement in energy mobilization and synthesis during the germination initiation phase (Fig 10).

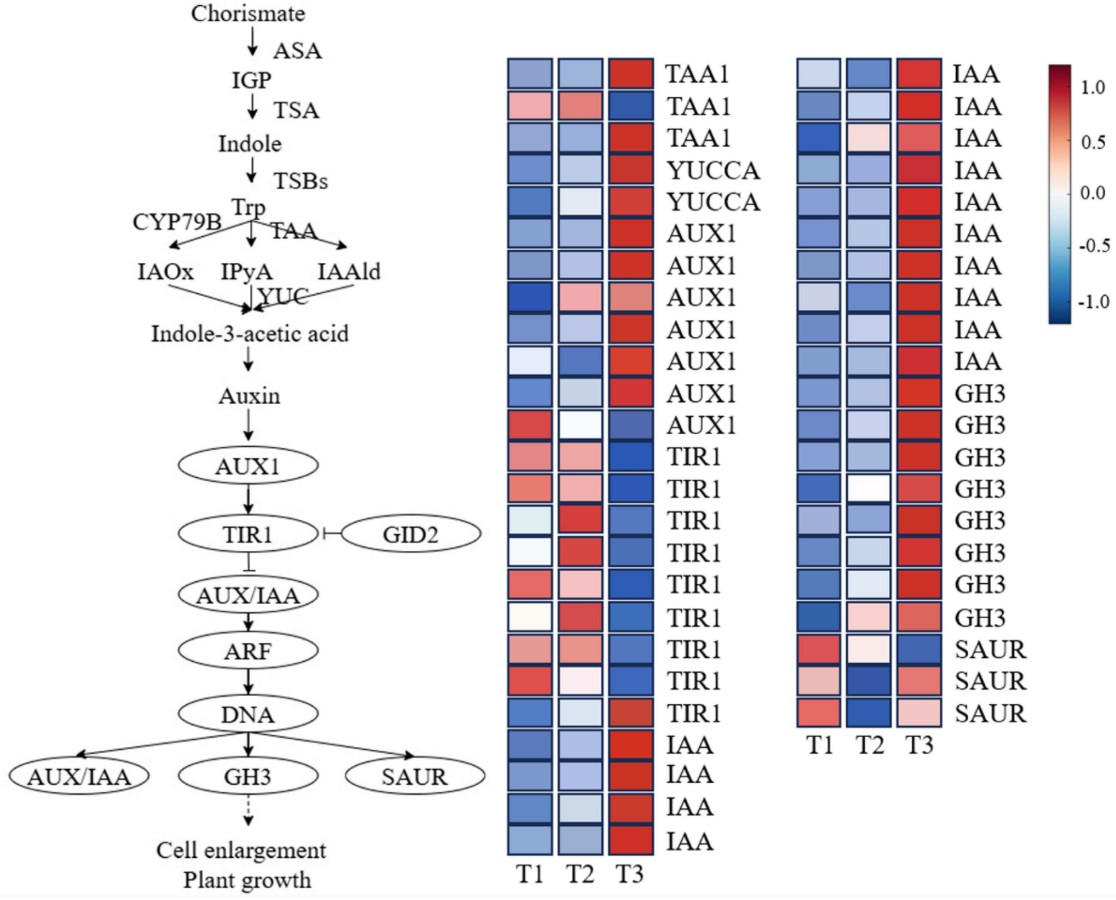

**Fig 9. Expression dynamics of key genes involved in auxin (IAA) biosynthesis and signal transduction during seed dormancy release in *Aconitum kusnezoffii*.**

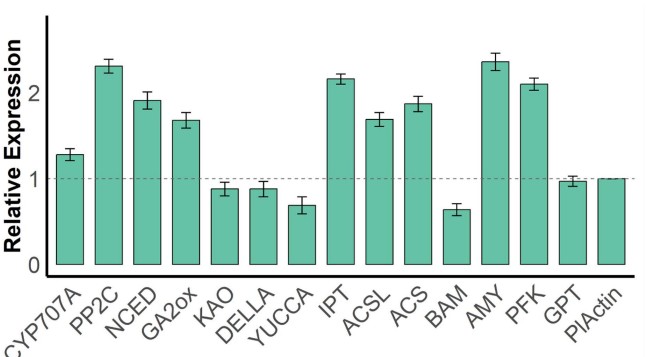

**Fig 10. Alidation of RNA-seq data by qRT-PCR for 14 selected genes involved in hormone signaling and metabolism in *Aconitum kusnezoffii*.**

Furthermore, genes related to regulation, such as *DELLA*, *ACS*, and *ACSL*, exhibited consistent expression patterns in both qRT-PCR and RNA-seq, further confirming the reliability of the sequencing data.

## Discussion

### Physiological significance of endogenous hormone dynamics and hormonal ratios during dormancy release in *Aconitum kusnezoffii*

The transition from seed dormancy to germination is governed by a complex network of endogenous hormones. The spatiotemporal dynamics and interactions among different hormones are central to determining the developmental fate of seeds [25]. In this study, the content changes of six major hormones—abscisic acid (ABA), gibberellins (bioactive GA, reported as $GA_3$-equivalents), indole-3-acetic acid (IAA), methyl jasmonate (MeJA), brassinolide (BL), and zeatin riboside (ZR; ELISA-based total tZR + cZR equivalents)—were systematically analyzed during seed dormancy release in *Aconitum kusnezoffii*. Additionally, key hormonal ratios ($GA_3$/ABA, IAA/ABA, and ZR/ABA) were evaluated. Results revealed clear stage-specific hormonal dynamics, with ratio trends closely associated with the progression of dormancy release.

In the early dormancy stage (T1), ABA levels were the highest, consistent with its well-established inhibitory role in maintaining seed dormancy [26,27]. As cold stratification proceeded, ABA levels significantly declined, reaching the lowest point at the radicle protrusion stage (T3), indicating de-repression of ABA signaling during dormancy release. Conversely, $GA_3$ and IAA levels increased rapidly from T2 to T3, displaying typical patterns of germination-promoting hormones. $GA_3$ is known to promote embryonic axis elongation by inducing hydrolytic enzyme activity and cell wall loosening [28], while IAA facilitates cell expansion and tissue polarity establishment [29]. In this study, both MeJA and BR also increased in the later stages, suggesting their potential involvement in dormancy release by modulating embryo stress responses and membrane system activation—a trend similarly reported in *Medicago sativa* [30] and *Lawsonia inermis* [31]. ZR content peaked during the imbibition stage, likely contributing to cell division activation [26].

Further analysis showed that the $GA_3$/ABA, IAA/ABA, and ZR/ABA ratios all increased markedly at T3, with IAA/ABA showing a continuous upward trend. These results suggest a phased reconstruction of antagonistic hormonal interactions. Notably, an increase in the $GA_3$/ABA ratio is widely recognized as a crucial physiological switch for breaking dormancy [32]. Meanwhile, the rising IAA/ABA and ZR/ABA ratios reflect the increasing dominance of growth-related signals such as cell expansion and division, consistent with observations in *Panicum virgatum* [33], *Cucumis sativus* [34], and *Leymus chinensis* [35].

### Expression patterns and regulatory mechanisms of hormone-related genes during dormancy release

The dynamic construction of hormone signaling networks plays a central role in regulating seed dormancy release in plants. This regulation involves not only the transcriptional control of biosynthetic and catabolic enzymes, but also the perception modules mediated by receptors, kinase cascade activations, and finely tuned regulation of response transcription factors. In this study, we focused on three representative phytohormones—ABA, GA, and IAA—to systematically analyze the expression dynamics of key genes and to elucidate the possible regulatory mechanisms during dormancy-to-germination transition in *Aconitum kusnezoffii*.

In the ABA biosynthesis pathway, the rate-limiting enzyme *NCED* showed high expression in stage T1, followed by significant downregulation, suggesting that ABA biosynthesis is primarily active in early dormancy. Conversely, the catabolic enzyme *CYP707A* was upregulated at T3, in line with the physiological data showing a sharp decline in ABA levels. These findings indicate that ABA depletion during dormancy release in *A. kusnezoffii* depends more on catabolism than on repressed biosynthesis [36].

In the ABA signaling pathway, the classic *PYL–PP2C–SnRK2–ABF* module exhibited a complex regulatory pattern. High expression of *PYL* at T1, along with elevated ABA concentration, contributed to the suppression of *PP2C* and *SnRK2*

activity. At T3, as ABA levels declined, both *PP2C* and *SnRK2* were upregulated, suggesting a feedback mechanism that releases ABA-mediated inhibition, thereby allowing the transition to other hormonal responses [37]. The downregulation of *ABF* at T3 reflects the termination of ABA signal output, paving the way for germination-promoting signals.

The GA pathway exhibited a more distinct "switch-like" regulatory pattern. Upstream biosynthetic enzyme *KAO* was continuously upregulated from T1 to T3, forming an antagonistic dynamic against declining ABA levels [38]. Meanwhile, the GA-deactivating enzyme *GA2ox* was significantly downregulated throughout, enhancing the biological activity of GA. At the signaling level, the *GID1–DELLA* module was distinctly regulated: *GID1* and *GID2* were upregulated at T3, while *DELLA* was downregulated, indicating enhanced GA signaling. This likely relieved DELLA-mediated repression and activated the transcription of genes involved in hydrolytic enzyme production and cell elongation, thereby promoting radicle protrusion [39].

The IAA-related pathway displayed a stepwise activation pattern, involving biosynthesis, transport, and signaling. Upregulation of *TAA1* and *YUCCA* facilitated IAA accumulation, and high expression of *AUX1* suggested enhanced polar auxin transport. The expression changes of *GH3* and *SAUR* genes indicated full activation of the auxin-responsive signaling module. Notably, the combined expression pattern of downregulated *TIR1* and upregulated *AUX/IAA* and *GH3* is regarded as a molecular hallmark of the transition from IAA perception to response activation [40]. The enhanced IAA signaling in *A. kusnezoffii* seeds may be associated with its capacity for shoot bud regeneration, which is consistent with the role of IAA in promoting dedifferentiation and callus induction in medicinal plants [41].

From a systems perspective, these three hormonal pathways displayed a competitive and coordinated interplay. ABA signaling was dominant during early dormancy (T1), but was gradually suppressed with the sequential activation of GA and IAA pathways. The downregulation of *PYL* and *ABF* reflects this transition, allowing GA and IAA to activate genes involved in growth and expansion, creating conditions for cell division and embryo axis elongation. This phenomenon, often referred to as the "hormonal switch window," has also been reported in several cold-region herbaceous species [42,43].

Moreover, multiple signaling elements such as *PP2C*, *SnRK2*, *TIR1*, and *GH3* exhibited feedback-responsive changes across different stages, suggesting that dormancy release is not a linear activation process. Rather, it involves dynamic feedback and positive regulatory loops between receptors, transcription factors, and biosynthetic enzymes. Such a complex signal topology enhances the robustness of regulation and enables flexible integration of external environmental cues.

The deep dormancy of *Aconitum kusnezoffii* seeds constrains large-scale cultivation and increases reliance on wild collection. By resolving the hormonal and transcriptional programs underlying dormancy release, our results provide actionable guidance for propagation and conservation. Practically, manipulating the GA/ABA and IAA/ABA balance via cold stratification regimes (e.g., alternating subzero/low temperatures) and, where permitted, cautious pre-sowing treatments (GA-based priming or low-dose auxin seed soaks) may accelerate transition to germination competence. Scheduling sowing to coincide with declining ABA and rising GA/IAA windows, combined with improved moisture management, can enhance uniform emergence in nurseries. Ecologically, fostering nursery-grown seedlings reduces pressure on wild populations and supports restoration initiatives in cold, high-altitude habitats where this perennial medicinal species contributes to biodiversity and rural livelihoods.

## Conclusions

This study systematically investigated the dynamic changes in embryo development, hormone metabolism, and transcriptional regulation during seed dormancy release in *Aconitum kusnezoffii*. Measurements of embryo development rate revealed that the embryo remained morphologically dormant during the first 14 days of cold stratification. A significant increase in embryo rate was observed after day 21, showing a sigmoidal growth pattern that clearly delineated three developmental stages: dormancy maintenance, dormancy initiation, and germination preparation. These findings provided

a reliable basis for selecting key sampling time points (0, 14, and 42 days) for subsequent hormone and transcriptome analyses.

Hormone profiling indicated that abscisic acid (ABA) was highly accumulated during early dormancy and then gradually declined, whereas the levels of germination-promoting hormones such as gibberellin (GA$_3$), indole-3-acetic acid (IAA), and methyl jasmonate (MeJA) increased significantly during the imbibition and radicle protrusion stages. The ratios of GA$_3$/ABA, IAA/ABA, and ZR/ABA also increased accordingly, suggesting that seed dormancy release is mediated through the synergistic action of multiple phytohormones.

Transcriptome sequencing identified 79,251 Unigenes, with functional annotations covering multiple public databases, confirming the high quality of sequencing data. Differential expression analysis showed that the transcriptional shift was most pronounced in the T1_vs_T3 comparison, which identified 38,250 differentially expressed genes (DEGs) enriched in pathways related to energy metabolism, hormone signaling, and RNA processing. Temporal expression clustering via STEM revealed multiple significant expression modules, reflecting stage-specific intrinsic regulatory transitions.

In hormone signaling pathways, *NCED* (ABA biosynthesis) was highly expressed at T1, while the ABA catabolic gene *CYP707A* was significantly upregulated at T3. In the signaling cascade, *SnRK2* was continuously upregulated, whereas *ABF* was downregulated at the later stage. In the GA pathway, *KAO* was consistently upregulated, *GA2ox* was downregulated, and the receptor components *GID1*/*GID2* were upregulated while *DELLA* was downregulated, collectively indicating enhanced GA signal output. For the IAA pathway, key biosynthetic and signaling genes including *TAA1*, *YUCCA*, *GH3*, and *SAUR* were significantly upregulated in the later stages, suggesting an increased role for IAA in promoting cell division and embryonic axis elongation.

In conclusion, the dormancy release process in *A. kusnezoffii* seeds is a hormone-coordinated, energy-restructuring, and transcriptionally dynamic event, orchestrated by complex interactions among multiple signaling pathways and metabolic regulators.

## Supporting information

**S1 File. Annotation.**
(XLS)

**S2 File. Counts.**
(XLS)

## Author contributions

**Conceptualization:** Yingtong Mu, Xiaojie Li, Xiaoming Zhang.

**Data curation:** Yingtong Mu, Xiaojie Li, Xiaoming Zhang.

**Formal analysis:** Yingtong Mu, Xiaojie Li, Xiaoming Zhang.

**Funding acquisition:** Xiaojie Li.

**Investigation:** Xiaojie Li.

**Methodology:** Kefan Cao.

**Resources:** Kefan Cao.

**Software:** Kefan Cao.

**Visualization:** Xiaojie Li.

**Writing – original draft:** Xiaojie Li.

**Writing – review & editing:** Xiaojie Li.

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
