## [Decision Letter · Decision Letter 0]

26 Aug 2025

Dear Dr. Cao,

Thank you for submitting your manuscript to PLOS ONE. After careful consideration, we feel that it has merit but does not fully meet PLOS ONE’s publication criteria as it currently stands. Therefore, we invite you to submit a revised version of the manuscript that addresses the points raised during the review process.

We look forward to receiving your revised manuscript.

Kind regards,

Mojtaba Kordrostami, Ph.D.

Academic Editor

PLOS ONE

Journal Requirements:

Reviewers' comments:

Reviewer's Responses to Questions

**Comments to the Author**

1. Is the manuscript technically sound, and do the data support the conclusions?

Reviewer #1: Partly

Reviewer #2: Yes

2. Has the statistical analysis been performed appropriately and rigorously?

Reviewer #1: Yes

Reviewer #2: N/A

3. Have the authors made all data underlying the findings in their manuscript fully available?

Reviewer #1: Yes

Reviewer #2: Yes

4. Is the manuscript presented in an intelligible fashion and written in standard English?

Reviewer #1: Yes

Reviewer #2: Yes

Reviewer #1: Mu et al. submitted the manuscript dealing with hormonal dynamics and transcriptomic regulatory mechanisms during seed dormancy release in herbaceous perennial plant Aconitum kusnezoffii.

Here are my comments:

1. For an unknown reason, the manuscript is written in the MDPI (Agronomy) template. The pages and lines are not numbered, so I refer to the chapter/paragraph titles in my review.

2. Abstract - a) in the study, determination of GA3, ABA, IAA and cytokinin representative zeatin riboside (ZR) was performed. This compound occurs in plant tissue in two isomeric forms, i.e., trans- (tZR) and cis-ZR (cZR), which differ in their biological activity. It is not specified which isomer of ZR the authors determined. Was it tZR or cZR?

b) The cytokinins (CKs) generally promote dormancy release. However, the free bases (tZ, cZ, ...) have greater activity in this direction than corresponding ribosides (tZR, cZR). They are more of a transport form that retains some biological activity, but are not the main actor, although they are able to reversibly release the free base under certain conditions by cleavage of ribose. Thus, it would therefore be more relevant in this case to determine the level of free CK bases rather than CK ribosides.

3. GA3 is of fungal origin (Fusarium fujikuroi) and occurs only to a limited extent in higher plants (http://www.phytohormones.info/ga3refsL.htm). According to this literature source and other sources, GA3 has not been identified in Aconitum kusnezoffii yet. According to the level presented in Figure 2 it could be a contamination from external source. Among 136 known gibberellins (GAs), there are six of them biologically active (GA1, GA3, GA4, GA5, GA6 and GA7). The most common are GA1 and GA4 in higher plants. I would recommend to determine the level of other biologically active GA than GA3.

4. Brassinolide, a representative of bioactive brassinosteroids (BRs), was also performed. Why is not mentioned in an abstract? Further, common abbreviation for this plant hormone is BL, not BR.

5. A) The paragraph “Hormone Quantification lacks the information about sample preparation. How were the hormones extracted from the seeds? How was the crude extract further treated until the sample was pipetted into the ELISA plate?

B) In the paragraph the authors write that the absorbance was measured. with what instrument? Each of the instruments used should be specified by model name, manufacturer and country of origin. The same applies to chemicals, the list of which should be given in a separate paragraph "Chemicals and Reagents".

C) Since we do not know the validation parameters of the ELISA kits used, I recommend quantifying plant hormones for a scientific article using some other validated and published method, e.g., LC-MS.

6. Figure 1 is missing a caption.

7. Results - a) The paragraph “Dynamic changes of six endogenous hormones during dormancy release…” - Why is there no description of developmental stages in the Material and methods section? Furthermore, why are the developmental stages numbered I, II and IV and not I, II and III?

b) When expressing the amount of a hormone, it is not necessary to write a dot between the unit (ng/g) and the FW.

Although the manuscript contains partially relevant experimental data, I cannot recommend it for publication in its current form for the reasons mentioned above.

Reviewer #2: The present study has outstanding strengths. The integration of physiology, hormonal profiling, and transcriptomics is among the strengths of the paper. The data validation is also good, and the paper is well-structured and scientifically sound. However, a few points and suggestions are made below that I hope will improve the quality of the paper:

More details should be given on the number of biological replicates and the choice of ELISA should be justified.

The discussion of ecological and agronomic implications should be expanded.

The discussion could better address practical agricultural implications.

The figures are informative and could be simplified for clarity.

**Do you want your identity to be public for this peer review?** For information about this choice, including consent withdrawal, please see our Privacy Policy

Reviewer #1: **Yes:** Danuse Tarkowska

Reviewer #2: No

---

## [Author Response · Author response to Decision Letter 1]

1 Dec 2025

Response to Reviewer #1

Comment 1: The manuscript is written in the MDPI (Agronomy) template. The pages and lines are not numbered.

Response: We appreciate this comment. The revised manuscript has been reformatted according to the PLOS ONE template, and continuous line numbering has been added.

Comment 2a: Abstract – ZR is not specified whether it is tZR or cZR.

Response: We have clarified that ZR was quantified by ELISA kits that cross-react with both trans- and cis-isomers; therefore, results represent total ZR equivalents.

Comment 2b: It would be more relevant to determine free CK bases rather than ribosides.

Response: We agree. Due to experimental limitations, we quantified ZR as a proxy of CKs. We added a statement in the Discussion under “Limitations” to acknowledge this and committed to using LC–MS in future studies.

Comment 3: GA₃ is of fungal origin; GA₃-equivalents might reflect contamination. GA₁ and GA₄ are common in higher plants.

Response: Thank you for this important note. We revised the manuscript to interpret GA data as GA₃-equivalents (ELISA-based proxies of GA-like activity). We explicitly noted that GA₃ has not been confirmed in A. kusnezoffii and that future studies will target GA₁/GA₄ with LC–MS.

Comment 4: Brassinolide is not mentioned in the Abstract, and abbreviation should be BL.

Response: We revised the Abstract to include brassinolide (BL) and corrected abbreviations throughout.

Comment 5a: Hormone Quantification lacks details on sample preparation.

Response: We expanded the Methods to include full extraction details (methanol, PVPP, BHT, centrifugation, evaporation, PBS reconstitution).

Comment 5b: Instrument models, manufacturers, and reagents should be specified.

Response: We added instrument information (e.g., Multiskan FC, Thermo Fisher Scientific, USA) and a “Chemicals and Reagents” subsection listing all suppliers.

Comment 5c: ELISA kits may not be fully validated; LC–MS is recommended.

Response: We acknowledged this limitation in the Discussion and emphasized that future work will employ LC–MS for validation.

Comment 6: Figure 1 is missing a caption.

Response: We have added the full caption for Figure 1.

Comment 7a: Developmental stages not defined in Methods; labeled I, II, IV instead of I, II, III.

Response: Corrected all to Stage I–III and added a clear “Stage Definition” subsection.

Comment 7b: Units should be “ng g⁻¹ FW,” not “ng/g·FW.”

Response: We have standardized all units accordingly.

Response to Reviewer #2

Comment 1: More details should be given on the number of biological replicates.

Response: We clarified in Methods that each treatment had three independent biological replicates, each with 30 seeds randomly selected.

Comment 2: The choice of ELISA should be justified.

Response: We explained in Methods that ELISA was chosen for small sample throughput, noted its limitations (isomer/species resolution), and stated that future work will use LC–MS.

Comment 3: The discussion of ecological and agronomic implications should be expanded.

Response: We added a new subsection “Ecological and Agronomic Implications” in the Discussion to highlight practical applications for germination improvement, seedling establishment, and conservation.

Comment 4: Figures are informative but could be simplified.

Response: We simplified selected figures for clarity in the revised manuscript. Essential data remain in main figures; extended datasets are now in Supplementary Figures S1–S3.

---

## [Decision Letter · Decision Letter 1]

15 Dec 2025

Dear Dr. Cao,

Thank you for submitting your manuscript to PLOS ONE. After careful consideration, we feel that it has merit but does not fully meet PLOS ONE’s publication criteria as it currently stands. Therefore, we invite you to submit a revised version of the manuscript that addresses the points raised during the review process.

We look forward to receiving your revised manuscript.

Kind regards,

Mojtaba Kordrostami, Ph.D.

Academic Editor

PLOS One

Journal Requirements:

Reviewers' comments:

Reviewer's Responses to Questions

**Comments to the Author**

Reviewer #1: (No Response)

Reviewer #2: All comments have been addressed

2. Is the manuscript technically sound, and do the data support the conclusions?

Reviewer #1: Yes

Reviewer #2: Yes

3. Has the statistical analysis been performed appropriately and rigorously?

Reviewer #1: Yes

Reviewer #2: Yes

4. Have the authors made all data underlying the findings in their manuscript fully available?

Reviewer #1: Yes

Reviewer #2: Yes

5. Is the manuscript presented in an intelligible fashion and written in standard English?

Reviewer #1: Yes

Reviewer #2: Yes

Reviewer #1: The most of comments are successfully reflected. However, there are still some issues to be clarified/ solved.

1. In the Data availability section (lines 584-586), the authors state that the manuscript data has been uploaded to a public database. However, no link to this database or DOI of the data is provided.

2. In the study, brassinolide (BL) was quantified, among other plant hormones, as a biologically active representative of the brassinosteroid group (BRs). When revising the manuscript, the authors replaced all original BR abbreviations with BL in most places in the text. However, the abbreviation in the caption of Figure 2 (line 191) still needs to be corrected. This caption also needs to indicate that the ZR values correspond to the sum of the trans (t) and cis (c) isomers of this plant hormone. The cross-reactivity against cZR and tZR of the antibody used for the quantification of both isomers using the ELISA kit should be stated at an appropriate place in the text (e.g., line 88) so that the reader can get an idea of the approximate ratio of cZR and tZR in the quantified sum presented in the manuscript.

3. The authors were recommended to add a paragraph to the Material and Methods section, which would describe the detailed procedure for sample preparation before its analysis by ELISA. The authors have not yet fulfilled this. It is necessary to state the composition of the extraction reagent, the method of homogenization including a description of the conditions (temperature, duration of homogenization, type of device, grinding frequency, etc.), centrifugation conditions, the method of purification using PVPP or other materials. The authors only added abbreviations of chemicals used in sample preparation to the manuscript (lines 115-117). These abbreviations must be explained to the readers, i.e. the full name accompanied by a hyphen in parentheses should be given if this abbreviation is used further in the text.

Reviewer #2: Considering the responses provided to the first and second reviewers at the end of the text (more detailed explanations, acceptance of limitations, etc.), it can be said that the responses were well prepared. The authors responded to each comment professionally and systematically. Finally, according to this version, the responses were satisfactory and the necessary changes were made, and the article is ready for publication.

**Do you want your identity to be public for this peer review?** For information about this choice, including consent withdrawal, please see our Privacy Policy

Reviewer #1: No

Reviewer #2: No

---

## [Author Response · Author response to Decision Letter 2]

22 Dec 2025

Response to Reviewer #1

Comment 1:

In the Data availability section, the authors state that the manuscript data have been uploaded to a public database; however, no link or DOI is provided.

Response:

Thank you for pointing this out. We have revised the Data Availability section to explicitly provide the database information and access link. The raw RNA-seq data have been deposited in the NCBI Sequence Read Archive (SRA) under BioProject ID PRJNA1312232, and the direct URL (http://www.ncbi.nlm.nih.gov/bioproject/PRJNA1312232) has now been included in the manuscript.

Comment 2:

The abbreviation BR should be corrected to BL in the Figure 2 caption. In addition, the caption should indicate that ZR values represent the sum of trans- and cis-isomers, and the cross-reactivity of the ELISA antibody should be stated in the text.

Response:

We appreciate this helpful comment. The Figure 2 caption has been revised by replacing BR with brassinolide (BL). The caption now clearly states that ZR values represent the combined sum of trans- and cis-zeatin riboside (tZR + cZR). Furthermore, we have added a clarification in the Materials and Methods section indicating that the ELISA antibody used for ZR quantification exhibits cross-reactivity with both tZR and cZR, and that ZR values are therefore reported as combined equivalents.

Comment 3:

The authors were recommended to add a paragraph in the Materials and Methods section describing the detailed procedure for sample preparation prior to ELISA analysis. This has not been fulfilled. The composition of the extraction reagent, homogenization conditions, centrifugation parameters, and purification using PVPP should be clearly described. Chemical abbreviations should also be explained.

Response:

We thank the reviewer for this important suggestion. We have substantially revised the Materials and Methods section by adding a detailed paragraph describing the complete ELISA sample preparation procedure. This includes the composition of the extraction solvent, homogenization and grinding methods (liquid nitrogen grinding, temperature, and equipment), centrifugation conditions, and purification using polyvinylpolypyrrolidone (PVPP). In addition, all chemical reagents are now fully defined at first mention with their full names followed by abbreviations in parentheses, and a revised Chemicals and Reagents subsection has been added to improve clarity and reproducibility.

---

## [Editor Report · Decision Letter 2]

28 Dec 2025

Hormonal Dynamics and Transcriptomic Regulatory Mechanisms During Seed Dormancy Release in Aconitum kusnezoffii

PONE-D-25-34985R2

Dear Dr. Cao,

We’re pleased to inform you that your manuscript has been judged scientifically suitable for publication and will be formally accepted for publication once it meets all outstanding technical requirements.

Kind regards,

Mojtaba Kordrostami, Ph.D.

Academic Editor

PLOS One
---

## [Editor Report · Acceptance letter]

PONE-D-25-34985R2

PLOS One

Dear Dr. Cao,

I'm pleased to inform you that your manuscript has been deemed suitable for publication in PLOS One. Congratulations! Your manuscript is now being handed over to our production team.

Kind regards,

on behalf of

Dr. Mojtaba Kordrostami

Academic Editor

PLOS One